# PVRIG Expression Is an Independent Prognostic Factor and a New Potential Target for Immunotherapy in Hepatocellular Carcinoma

**DOI:** 10.3390/cancers15020447

**Published:** 2023-01-10

**Authors:** David Jeremie Birnbaum, Maelle Picard, Quentin Da Costa, Thomas Delayre, Pascal Finetti, Olivier Cabaud, Emilie Agavnian, Bernadette De Rauglaudre, Emilie Denicolaï, François Bertucci, Emilie Mamessier

**Affiliations:** 1Predictive Oncology Laboratory, Labeled Team Ligue Nationale Contre Le Cancer, Centre de Recherche en Cancérologie de Marseille (CRCM), Paoli-Calmettes Institute (IPC), Inserm UMR1068, CNRS UMR7258, Aix-Marseille University, 13009 Marseilles, France; 2General and Visceral Surgery Department, North’s Hospital, 13015 Marseilles, France; 3IPC/CRCM Experimental Pathology (ICEP), CRCM, Paoli-Calmettes Institute, 13009 Marseilles, France; 4Department of Digestive Oncology and Gastro-Enterology, Timone Hospital, AP-HM, 13005 Marseille, France; 5Department of Medical Oncology, Paoli-Calmettes Institute, 13009 Marseilles, France

**Keywords:** hepatocellular carcinoma, immunotherapy, *PVRIG*, CD112R, survival

## Abstract

**Simple Summary:**

Hepatocellular carcinoma (HCC) represents the third cause of cancer-related death in the world, and identification of new prognostic factors and/or therapeutic targets remains a major issue. Recently, immune checkpoint inhibitors (ICIs) have opened promising therapeutic options for this disease. Using a large series of transcriptomic data of clinically annotated HCC samples, we provide evidence that the TIGIT/DNAM-1 axis, and notably the PVRIG molecule, might be an interesting therapeutic candidate for HCC tumors.

**Abstract:**

Hepatocellular carcinoma (HCC) is a frequent and deadly cancer in need of new treatments. Immunotherapy has shown promising results in several solid tumors. The TIGIT/DNAM-1 axis gathers targets for new immune checkpoint inhibitors (ICIs). Here, we aimed at highlighting the potential of this axis as a new therapeutic option for HCC. For this, we built a large transcriptomic database of 683 HCC samples, clinically annotated, and 319 normal liver tissues. We interrogated this database for the transcriptomic expression of each member of the TIGIT/DNAM-1 axis and tested their prognostic value for survival. We then focused on the most discriminant one for these criteria, i.e., *PVRIG*, and analyzed the clinical characteristics, the disease-free and overall survivals, and biological pathways associated with *PVRIG* High tumors. Among all members of the TIGIT/DNAM-1 axis, *PVRIG* expression was higher in tumors than in normal liver, was heterogeneous across tumors, and was the only member with independent prognostic value for better survival. *PVRIG* High tumors were characterized by a higher lymphocytic infiltrate and enriched for signatures associated with tertiary lymphoid structures and better anti-tumor immune response. These results suggest that patients with *PVRIG* High tumors might be good candidates for immune therapy involving ICIs, notably ICIs targeting the TIGIT/DNAM-1 axis. Further functional and clinical validation is urgently required.

## 1. Introduction

Hepatocellular carcinoma (HCC) represents the third cause of cancer-related death in the world [1,2,3]. Its incidence is increasing [4,5], with one million of new cases expected worldwide in the upcoming years. HCC represents a heterogeneous disease with multiple causal factors (alcohol intake, infection with hepatitis B (HBV) or hepatitis C (HCV) viruses, and metabolic syndrome) and miscellaneous prognoses [6,7,8]. Surgical resection, local ablation, and liver transplantation are the only curative treatments selected for patients and confer a five-year survival rate above 50%. Trans-arterial chemoembolization is used in the palliative setting, for patients with advanced disease (with a 20% improvement on the two-year overall survival (OS) rate) [9]. In case of failure, or if the patient is not eligible for trans-arterial chemoembolization, sorafenib and lenvatinib, two multi-kinase inhibitors, remain the only Food and Drug Administration (FDA)-approved drugs [10]. Median OS for these patients reaches 6.2 and 10.9 months in the Asia-Pacific region and Western countries, respectively [11,12]. Hence, as for many aggressive cancers, there is an urgent need for new therapeutic options [13].

Immunotherapies are part of these new therapeutic options in solid tumors [14]. They mainly rely on immune checkpoint inhibitors (ICIs) directed against CTLA-4, PD-1 or PD-L1 [15]. A response is observed in a variable proportion of patients with aggressive tumors (around 10 to 20%) who previously had no other option. ICIs are thus considered as major assets in the treatment of several aggressive cancers, such as triple-negative breast cancer, lung, melanoma, and HCC [16]. For now, the treatment of HCC with nivolumab or pembrolizumab, two anti-PD-1 ICIs, has shown prolonged median OS and progression-free survival (PFS) in a subset of patients [17]. However, the FDA has recently emitted accelerated approval to nivolumab (an anti-PD-1) and to the association of nivolumab and ipilimumab (an anti-CTLA-4) in the second line setting, for patients who are progressing on sorafenib. This decision was based on the results of the phase I/II study (NCT01658878) that shows a better overall response rate and duration of response in patients treated with these two ICIs [18,19]. No approval has however been granted by European institutions for immunotherapies in HCC yet. Most importantly, these studies highlight the fact that HCCs are tumors sensitive to immune therapy. In this regard, efforts are being made to identify other ICIs as alternative targets for cancer immunotherapies.

In this line, the TIGIT/DNAM-1 axis might be an option for the next generation of cancer immunotherapies [20]. This axis is composed of nine members, including five receptors, namely DNAM-1 (CD226, activator), TACTILE (CD96, activator), TIGIT (inhibitor), PVRIG (CD112R, inhibitor), and KIR2DL5A (CD158f, inhibitor), all expressed on cytotoxic cells (CD8^+^ T cells, Natural killer (NK) cells, and γδ T cells), and four ligands, namely NECTIN-1 (PVRL1/CD111), NECTIN-2 (PVRL2/CD112), NECTIN-3 (PVRL3/CD113), and PVR (CD155), which can be expressed on antigen-presenting cells or tumor cells [21]. Even though studies putting forward the role of TIGIT/DNAM as a regulator of HCC are still particularly scarce in the literature, one recent study reports that the blockade of TIGIT enhanced the antitumor activity of CD8^+^ T cells during the progression of HBV-related HCC in a spontaneous HCC mouse model. Interestingly, the blockade of PD-L1 did not slow HCC growth in HBs-HepR mice when administered alone, nor synergized with TIGIT blockade. This recent finding provides robust data indicating that the TIGIT/DNAM-1 axis might be a better candidate than the PD-1/PD-L1 axis in HCC (at least in HBV-related HCC) [22]. Additional examples along this line can be found for other tumor types. Indeed, in murine colon carcinoma models, combined blockade of TIGIT and PD-1, has shown synergic enhancement of CD8^+^ T cell function [23]. In 2019, seven clinical trials have used the anti-TIGIT alone or in combination [24]. The anti-TIGIT is certainly the most advanced ICIs issued from the TIGIT/DNAM1 axis, but others are in development, notably against the second inhibitory receptor of the axis, PVRIG. In vivo studies on acute myeloid leukemia and melanoma revealed that PVR blockade led to anti-tumoral activation and prevented metastasis formation [25]. In humans, simultaneous blockade of PVRIG and TIGIT enhanced trastuzumab-mediated NK cell response against breast cancers [26]. There is currently one anti-PVRIG antibody, which is currently tested in an ongoing clinical trial, which is either administered alone or in combination with nivolumab or anti-TIGIT ICI in several solid tumors (NCT03667716). Altogether, these clinical trials suggest that the TIGIT/DNAM-1 axis is indeed a target for the upcoming generation of ICIs, and a new therapeutic opportunity for the treatment of several tumor types, potentially including HCC [24].

Based on these observations, we analyzed the transcriptomic expression of the members of the TIGIT/DNAM-1 axis to search for correlations between mRNA expression and clinicopathological data, including survival. The overall objective was to assess their relevance as potential targets for new ICIs in HCC.

## 2. Materials and Methods

### 2.1. Gene Expression Datasets

We gathered clinicopathological and gene expression data of HCC clinical samples from public repositories: Gene Expression Omnibus (GSE14520 and GSE54236) and TCGA database (Appendix A). We collected 683 HCC samples and 319 healthy liver tissues (1002 samples in total) that we pooled in our database. Patients were from both Western and Eastern countries. The analysis’s workflow is detailed in Figure 1. Among the 683 HCC samples, the follow-up and survival data were available in two datasets (GSE14520 and TCGA), representing 602 HCC samples (Figure 1).

### 2.2. Gene Expression Data Processing

Data analysis required pre-analytic processing, as previously described [27]. Briefly, each DNA-microarray dataset was normalized separately, by using quantile normalization for the available processed Agilent data and Robust Multichip Average (RMA) with the non-parametric quantile algorithm for the raw Affymetrix dataset. Normalization was performed in R using Bioconductor and associated packages (version 3.5.1, R Foundation for Statistical Computing, Vienna, Austria). We log_2_-transformed the available TCGA RNA-seq data that were already normalized. Then, we mapped hybridization probes across the different technological platforms. We used EntrezGene (Homo sapiens gene information db, release date: 9 December 2008, ftp://ftp.ncbi.nlm.nih.gov/gene/) to retrieve and update the non-Affymetrix gene chips annotations, and NetAffx Annotation files (www.affymetrix.com; release date: 1 December 2008) for Affymetrix annotations. The probes were then mapped according to their EntrezGeneID. When multiple probes represented the same GeneID, we retained the one with the highest variance in a particular dataset. Next, we corrected the three studies for batch effects using z-score normalization. Briefly, for each gene expression value in each study separately, all values were transformed by subtracting the median of the gene in that dataset divided by its standard deviation, median, and standard deviation being measured on HCC samples only. The comparison of gene expression levels (continuous values) in HCC versus normal tissues (NT) was performed using the Student *t*-test. *PVRIG* expression in tumors was measured as a discrete value (High versus Low) by using the median expression level across the whole series as cut-off.

We applied several multigene signatures, including the molecular subtypes classification of Hoshida et al. [28], to each dataset separately. Because PVRIG is a potential target of immunotherapy, we searched for correlations between its expression in HCC and different immune variables. We thus tested several immune- and fibroblastic-multigene classifiers/scores: the tumor-infiltrating lymphocyte (TIL) score [29], the 24 Bindea’s innate and adaptive immune cell subpopulations signatures [30], the Immunologic Constant of Rejection (ICR) classifier [31], the T-cell-inflammation signature (TIS) [32], the tertiary lymphoid structures (TLS) signature [33], the cytolytic activity score [34], the antigen processing machinery (APM) score [35], and three different classifications for fibroblast subsets [36,37,38].

To explore the biological pathways associated with *PVRIG* expression in HCC, we performed a supervised analysis with the GSE14520 dataset [39] as learning set, which included 121 *PVRIG* High and 121 *PVRIG* Low samples. We used the TCGA dataset [40] as an independent validation set, which included 180 *PVRIG* High and 180 *PVRIG* Low samples. In the learning set, we compared the whole-transcriptome expression profiles between *PVRIG* High and *PVRIG* Low samples using a Student *t*-test and applied a false discovery rate (FDR) correction. Significant genes were defined by the following thresholds: adjusted *p*-value < 0.05 and fold change (FC) superior to |1.5x|. Ontology analysis of the resulting gene list was established with the GO biological processes of the Bioconductor package clusterProfiler. We tested the robustness of the resulting gene list in the validation set by computing the metagene score for *PVRIG* High samples and *PVRIG* Low samples, respectively.

### 2.3. Statistical Analysis

Continuous variables were displayed as median (range) values and categorical variables with counts (n) and proportion (%). Correlations between the *PVRIG* expression-based tumor groups (*PVRIG* High and Low) and clinicopathological and molecular variables were analyzed using the Student *t*-test or the Fisher’s exact test when appropriate. The disease-free survival (DFS) was calculated from the date of diagnosis until the date of relapse (local, regional, or distant) or death from any cause. OS was calculated from the date of diagnosis to the date of death from any cause. Follow-up was measured from the date of diagnosis to the date of last news for event-free patients. We compared DFS and OS of the tumor groups by using Kaplan–Meier curves and log-rank tests. Uni- and multivariate prognostic analyses for DFS and OS were based on the Cox regression analysis (Wald test) and searched for associations between the gene expression level (continuous value) of members of the TIGIT/DNAM-1 and PD-1/PD-L1 axes and DFS and OS, then between the *PVRIG* expression-based tumor groups and the clinicopathological variables and DFS and OS. To measure the association between the *PVRIG* expression-based tumor groups and immune- and fibroblast-related classifiers/scores, we used the glm R function based on a binomial model with logit link. The multivariate analyses were run using the variables significant in univariate analysis (*p* < 0.05). All statistical tests were two-sided at the 5% level of significance. Statistical analysis was performed using the survival package (version 2.42) in the R software (version 3.5.1; http://www.cran.r-project.org/, release date: 2 July 2018).

## 3. Results

### 3.1. HCC Patients’ Clinical Characteristics and PVRIG Expression

Clinicopathological data of the 683 tumors gathered in our database are summarized in Table 1. Briefly, the median value for patients’ age at diagnosis was 56 years (range (16–90) and ~76% of patients were male. Most of the samples were of stage I for the TNM staging (46%) [41], and grade 2 (48%) and grade 3 (34%) for the pathological grading. Serum AFP was predominantly below 300 ng/mL (55%). The majority of patients were positive for HBV infection (57%) and 16% were positive for HCV. Alcohol consumption was reported for 34% of patients, non-alcoholic fatty liver disease (NAFLD) for 6%, and hemochromatosis for 2%. According to the molecular subtyping of Hoshida [28], most tumors were classified S3 subtype (differentiated, 47%), then S1 (stromal, 31%), and S2 (stem, 22%) subtypes. In the entire cohort, the median DFS was 28 months (range (1–121)), with a 5-year DFS of 34% (95% CI (30–40)), and the median OS was 70 months (range (1–121)), with a 5-year OS of 53% (95% CI (48–58); Appendix A). The median follow-up was 52 months (95% CI (49–54)).

### 3.2. Transcriptomic Expression of Members of the TIGIT/DNAM-1 Axis in HCC

We first compared the gene expression level of each member of the TIGIT/DNAM-1 axis in HCC (N = 683) and in NT (N = 319) tissues: TIGIT, DNAM-1, TACTILE, PVRIG, and KIR2DL5A and their ligands PVR, NECTIN-1, NECTIN-2, and NECTIN-3 (Figure 2A). PD-1 and PD-L1, whose expression has already been studied in HCC, were added for comparison. TIGIT, DNAM-1, TACTILE, PVRIG, and KIR2DL5A expressions were lower (Student *t*-test) in HCC than in NT (*p* = 5.4 × 10^−3^, *p* = 2.2 × 10^−27^, *p* = 2.5 × 10^−15^, *p* = 6.6 × 10^−29^, and *p* = 9.5 × 10^−12^, respectively) (Figure 2B). Ligands from the TIGIT/DNAM-1 axis were variable in expression: expression of PVR and NECTIN-2 were higher in HCC than in NT (*p* = 7.1 × 10^−4^ and *p* = 1.3 × 10^−32^, respectively). As these ligands both interact with activating (DNAM-1) and inhibitory (TIGIT and PVRIG) receptors on cytotoxic cells, targeting TIGIT and PVRIG might promote the engagement of DNAM-1. Inversely, expression of NECTIN-1 and NECTIN-3 were lower in HCC than in NT (*p* = 1.9 × 10^−9^ and *p* = 1.5 × 10^−6^ respectively). PD-1 expression was lower in HCC than in NT (*p* = 5.0 × 10^−1^), whereas PD-L1 expression was not significantly different between HCC and NT. Overall, these results showed that, regarding the TIGIT/DNAM-1 axis, the microenvironment of HCC was mostly inhibitory for cytotoxic cells, with NECTIN-2 being the predominant ligand expressed in HCC tissues.

As a pre-screen strategy to identify which of these members was important to predict HCC outcome, we searched for a correlation between the gene expression of each member of the TIGIT/DNAM-1 axis and DFS and OS data. Gene expression was analyzed as a continuous variable. As shown in Figure 3, *PVRIG* and *NECTIN-1* were the only members of the TIGIT/DNAM-1 axis that were significantly associated with OS in uni- and multivariate analyses (in univariate analysis: *p* = 3.2 × 10^−6^ and *p* = 1.8 × 10^−2^, respectively; in multivariate analysis: *p* = 4.6 × 10^−4^ and *p* = 5.5 × 10^−3^, respectively; Wald test *p* = 2.7 × 10^−5^). *PVRIG* being also significantly associated with DFS, contrary to *NECTIN-1* (Appendix A), we decided to further focus our analysis on *PVRIG*.

### 3.3. PVRIG Gene Expression and Correlation with Clinico-Pathological Features and Molecular Subtypes Classification

*PVRIG* gene expression was heterogeneous across samples, with a range of intensities covering six intervals in log_2_ scale (Appendix A). We thus divided the population into two groups, based on *PVRIG’s* median expression level in the whole series, respectively, defining a *PVRIG* Low group and a *PVRIG* High group. We first searched for correlations between the clinicopathological variables and molecular subtypes classification and the *PVRIG* Low versus High gene expression status. As shown in Table 2, no difference was found with patients’ age, TNM stage, pathological grade, AFP level, HBV and HCV infection status, alcohol consumption, and hemochromatosis. By contrast, differences existed with a higher proportion of females (*p* = 2.9 × 10^−2^), with more cases positive for NAFLD (*p* = 3.6 × 10^−2^) and more Hoshida’s S1 subtypes (*p* = 1.9 × 10^−7^) in *PVRIG* High tumors (Table 2). Overall, there was no major clinical feature associated with *PVRIG* High tumors, except that these tumors were significantly enriched for the Hoshida’s S1 subtype, which is characterized by a fibrotic and immune-rich environment.

### 3.4. PVRIG Gene Expression and Correlation with Survival Data

We then searched for correlations between *PVRIG* Low and High groups and DFS data, available for 552 patients. In the whole population, the median DFS was 28 months (range (0–121)) and the 5-year DFS was 34% (95% CI (30–40)). Coherent with the uni- and multivariate analyses performed with the continuous values of *PVRIG*, *PVRIG* Low and High groups showed different DFS values: the median DFS was 40 months (range (0–121)) in the *PVRIG* High group and 19 months (range (0–114)) in the *PVRIG* Low group (*p* < 0.0001), with a higher 5-year DFS rate in *PVRIG* High than in *PVRIG* Low samples (*p* = 4.0 × 10^−5^) (Table 2, Figure 4A).

In univariate analysis, in addition to *PVRIG* Low and High groups (*p* = 5.1 × 10^−5^), the other variables associated with DFS were the TNM/UICC classification and the HBV infection status (*p* = 1.9 × 10^−3^ and *p* = 1.8 × 10^−6^, respectively) (Table 3). In multivariate analysis, *PVRIG* Low and High groups, the TNM/UICC classification, and the HBV infection status remained significant (Table 3). Thus, the *PVRIG* gene expression level is an independent prognostic factor for DFS in HCC patients (HR = 0.7, range [0.5–0.8], *p* = 8.5 × 10^−4^).

Similarly, we looked at how *PVRIG* Low or High expression influenced OS data, which were available for 602 patients. In the whole population, the median OS was 69.5 months (range [0–120.7]) and the 5-year OS was 52.7% (95% CI [47.8–58]). The median OS was 80.7 months (range [0–120.7]) in the *PVRIG* High group and 53.3 months (range [0–114.3]) in the *PVRIG* Low group (*p* < 0.001) with a higher 5-year OS rate in the *PVRIG* High group than in the *PVRIG* Low group (*p* = 1.2 × 10^−4^) (Table 2, Figure 4B).

In univariate analysis the variables associated with OS were the *PVRIG* groups (*p* = 1.4 × 10^−4^), the TNM/UICC classification (*p* = 1.5 × 10^−12^), HBV infection status (*p* = 1.5 × 10^−5^), and the Hoshida’s subtypes classification (*p* = 4.2 × 10^−4^) (Table 4). In multivariate analysis, all these variables remained significant, with the exception of the HBV infection status. This confirmed the independent prognostic value of *PVRIG* gene expression levels (HR = 0.6, range [0.4–0.8], *p* = 9.1 × 10^−4^) (Table 4).

### 3.5. PVRIG Expression and Correlation with Biological Processes

To investigate the biological functions associated with *PVRIG* in HCC pathogenesis, we applied a supervised analysis to search for the genes differentially expressed between the *PVRIG* Low and *PVRIG* High groups. We first worked with a learning set (GSE14520, n = 242 divided in *PVRIG* High (n = 121) and Low (n = 121) groups). The supervised analysis identified 181 differentially expressed genes, including 15 genes downregulated and 166 genes upregulated in the *PVRIG* High group (Figure 5A and Appendix A). The robustness of this gene signature was confirmed in the independent validation set (TCGA dataset), including a total of 360 HCC samples, by comparing the metagene score of *PVRIG* Low samples and *PVRIG* High samples using a Student *t*-test. As expected, the metagene’s difference was not only significant in the learning set but, more importantly, also in the 360 samples from the independent validation set (Figure 5B), showing its robustness.

The ontology analysis of the 15 genes upregulated in *PVRIG* Low tumors highlighted oxidation-reduction processes (*p* = 1.8 × 10^−2^), a bile acid biosynthetic process (*p* = 3.6 × 10^−2^), and a tricarboxylic acid cycle (*p* = 4.9 × 10^−2^). This number of genes was however very small, and these data should be interpreted with caution. Inversely, the functional enrichment analysis of the 166 genes upregulated in *PVRIG* High tumors revealed pathways related to an immune response (*p* = 1.6 × 10^−35^), a T cell co-stimulation (*p* = 1.1 × 10^−13^), an inflammatory response (*p* = 5.1 × 10^−13^), a chemokine-mediated signaling (*p* = 6.7 × 10^−13^), a signal transduction (*p* = 3.1 × 10^−12^), and a defense in response to virus (*p* = 3.6 × 10^−10^) (Figure 5C and Appendix A). Immune response and inflammation were the predominant pathways activated in *PVRIG* High tumor samples.

### 3.6. PVRIG Gene Expression Status and Correlation with Immune and Stromal Features

To explore further the quality of the immune response observed in *PVRIG* High tumors, we determined if there was a difference in the quantity of immune or stromal cells infiltrating *PVRIG* Low and *PVRIG* High tumors. For this we used the TILs score tool, which provides not only scores for tumor cells’ purity but also for stromal, lymphoid, and myeloid cells’ infiltration within the analyzed tumor. *PVRIG* High samples showed the highest score for lymphoid cells, followed by myeloid cells and stromal infiltrates (respectively, *p* = 1.1 × 10^−31^, *p* = 1.0 × 10^−19^, and *p* = 1.5 × 10^−6^). No difference was observed between *PVRIG* Low and *PVRIG* High tumors regarding tumor cells. This identified *PVRIG* High tumors as being more infiltrated by immune cells, notably lymphocytes, than *PVRIG* Low tumors (Figure 6A, and Appendix A).

We next looked at the detailed composition and functional orientation of these immune infiltrates. As expected from the ontology, the analysis of the 24 Bindea’s immune cell types, defined as the immunome [30], revealed that *PVRIG* High tumors differed from *PVRIG* Low tumors by a higher infiltration with immune cell populations (*p* < 0.05). The top fifteen subsets infiltrating *PVRIG* High tumors were the lymphocytes (T cells: *p* = 6.7 × 10^−31^; B cells: *p* = 3.4 × 10^−28^; Th1 cells: *p* = 4.3 × 10^−22^; γδT cells: *p* = 9.6 × 10^−22^; Tem cells: *p* = 1.2 × 10^−16^; TfH cells: *p* = 2.4 × 10^−16^; CD8 T cells: *p* = 9.1 × 10^−14^; Tcm: *p* = 6.3 × 10^−11^), the dendritic cells (aDC: *p* = 2.8 × 10^−19^; iDC: *p* = 1.9 × 10^−13^) and the macrophages (*p* = 4.5 × 10^−12^). Notably, *PVRIG* High tumors showed a higher infiltration with cytotoxic cell subsets (cytotoxic cells: *p* = 8.9 × 10^−28^; NK cells: *p* = 5.7 × 10^−14^; NK CD56dim cells: *p* = 6.8 × 10^−15^; NK CD56bright cells: *p* = 4.4 × 10^−11^) (Figure 6B and Appendix A). This profile was strongly supportive of the presence of an anti-tumor or anti-viral immune response from both innate and adaptive immune effector cells in *PVRIG* High tumors.

Additional immune functional signatures re-enforced this observation at the functional level, by showing a higher ICR score (*p* = 9.8 × 10^−27^) [31], TIS (*p* = 1.9 × 10^−26^) [32], TLS signature (*p* = 1.2 × 10^−25^) [33], and immune cytolytic activity score (*p* = 7.7 × 10^−25^) [34] in *PVRIG* High tumors. All these scores/signatures are biologically and clinically relevant of cytotoxic activity from adaptive immune effector cells [32,33,34,42]. Some of them, such as TIS and TLS, are also predictors of a response to ICIs [43,44,45]. Altogether, these data suggested that *PVRIG* High tumors might be good responders to ICIs, thanks to pre-infiltrated primed effectors cells. Of note, the antigen APM score [35] was also significantly enhanced (*p* = 2.6 × 10^−6^) and in favor of a mounted acquired anti-tumor or anti-viral immune response, but this signature was not among the top functional features associated with *PVRIG* High tumors (Figure 6C and Appendix A).

We then looked into the 166 genes upregulated in the *PVRIG* High tumors (Appendix A) and searched for signs of exhaustion, which would be coherent with tumor development despite cytotoxic infiltration and activity. Except for *IDO1*, none of the classical markers of exhaustion (*PDCD1/PD-1, ENTPD1, LAG3, HAVCR2/TIM3, CTLA4, CD160, 2B4, BTLA, TIGIT, TOX, EOMES,*…) were detected [46]. Inversely, the absence of key molecules associated with a highly efficient anti-tumor immune response (*IFNG*, *IL12A* and *IL12B*, *IL18*, *TNF*, *GNLY*, *KLRC2, KLRC3,…*) suggested that the cytotoxic potential of infiltrated immune cells was not totally unleashed despite expression of immune cell activation markers (*GZMA, GZMB, GZMK, PRF1, CD8A, NKG7, STAT1, IRF1, TARP, KLRK1, PTPRC, CCL5, IL2RG, IL2RB, LCK, PIK3CD, CD69, CD52, CD53, TAP1,*…) (Appendix A).

Finally, we looked at the composition of stromal cells, and more especially of CAF. We used the Givel et al., Kieffer et al., and Biffi et al. signatures to search for enrichment in specific CAF subsets. *PVRIG* High tumors were enriched with both CAF-S1 and CAF-S4 subsets (*p* = 1.5 × 10^−12^, *p* = 1.5 × 10^−9^, respectively) [47]. Similarly, both inflammatory CAF (iCAF) and myofibroblastic CAF (myCAF) subsets were detected in *PVRIG* High tumors. The iCAF subset was slightly predominant, especially the IFNy-iCAF (*p* = 6.3 × 10^−21^), the IL-iCAF (*p* = 6.3 × 10^−15^), and the detox-iCAF (*p* = 5.8 × 10^−11^) (Figure 6D and Appendix A). These subsets secrete immunomodulatory molecules that can elicit or maintain an anti-tumor immune response. The pro-tumoral myCAF subsets (wound-, ecm-, and TGFb-myCAFs) were also well represented (wound-myCAF: *p* = 1.4 × 10^−9,^ ecm-myCAF: *p* = 1.7 × 10^−8^, TGFβ-myCAF: *p* = 3 × 10^−8^). These subsets are pro-tumoral cells (Figure 6D).

Altogether, these results suggest that cytotoxic immune cells are infiltrating *PVRIG* High tumors, concomitant with a mixed anti-tumoral iCAF-like and pro-tumoral myCAF-like stroma.

## 4. Discussion

Based on the gene expression analysis of the TIGIT/DNAM-1 axis in a large cohort of HCC clinical samples, we showed that *PVRIG* is an independent biomarker predictive for favorable clinical outcome. *PVRIG* expression was lower in HCC samples than in normal hepatic tissues. Its expression was heterogeneous across tumors and allowed us to search for correlations between *PVRIG* expression levels and clinicopathological variables. Using *PVRIG* median as a cutoff enabled us to interrogate our database, avoiding time-consuming immunohistochemistry development and analyses, while providing opportunities to work on a relatively large series of samples. We were then able to search for association with the expression of other genes and multigene signatures to investigate pathways associated with *PVRIG* gene expression levels more in depth.

To our knowledge, this is the third study highlighting *PVRIG* expression as a new potential target for immune therapy in HCC. *PVRIG* High expression has indeed been associated with improved DFS and OS in HCC patients [48], and an eight-gene prognostic signature was constructed to anticipate the HCC outcome, in which *PVRIG* was the most important variable [49].

PVRIG is an immune checkpoint receptor with inhibitory function for cytotoxic CD8^+^ T cell and NK cells [50,51]. In murine models with solid tumors, PVRIG interaction with its ligand NECTIN-2 on tumor or dendritic cells surface suppresses cytotoxic signals, cytokines production, and triggers exhaustion [51,52].

To our knowledge, the correlation between *PVRIG* expression and immune variables has never been explored further, and this is the first report characterizing the tumor immune microenvironment and biological pathways in *PVRIG* High tumors. The HBV and HCV infection status being equally distributed in *PVRIG* High or Low tumors, the background immune response due to the viral infection can be considered as equivalent. Compared to *PVRIG* Low tumors, several immune variables were enriched in *PVRIG* High tumors and suggested: (i) higher infiltration of immune cells; (ii) higher infiltration of both adaptive and innate immune cells, notably B cells, γδ T cells, T cells with a Th_1_ profile, cytotoxic CD8^+^ T cells, and activated NK CD56dim cells; and (iii) higher cytotoxic activity. Altogether, these results suggested that *PVRIG* High tumors have a higher potential for anti-tumor immune response than *PVRIG* Low tumors. Nonetheless, despite this apparent anti-tumoral environment, the local immune cells did not prevent HCC occurrence and progression. The high expression of *NECTIN-2,* the ligand of *PVRIG,* which has also been reported by others in HCC tumors [53], might bridle anti-tumor response efficiency. Indeed, we did not find major evidence for exhaustion, as often observed in highly immunogenic tumors [27,54,55]. The higher infiltration with both immunosuppressive and pro-tumoral CAF subsets in *PVRIG* High tumors was also in line with the hypothesis of a poor anti-tumoral response.

An apparent contradiction in our results lies in the association of *PVRIG* High expression, which can trigger a strong immunosuppressive signal, with better disease-free and overall survivals. We and others have previously reported similar results with expression of *PD-L1* [55,56,57] and *IDO1* inhibitory molecules [58]. In these cases, our explanation is that a very strong anti-tumoral immune response, associated with better clinical outcome, also activates an inhibitory feedback loop, in the prospect of a return to homeostasis with minimum tissue damages. IFN-γ was in part responsible for the overexpression of PD-L1 and the initiation of the negative feedback loop in these tumors [53]. Here, we did not report major signs of exhaustion, except for *IDO1*, and we did not detect strong IFN-γ production either, but there was evidence for general IFN and cytotoxic immune responses. One hypothesis might be that the PVRIG/NECTIN-2 axis is a new negative feedback loop, either proper to the tumor liver microenvironment or associated with a specific context of immune response. Indeed, the TIGIT-DNAM-1 axis, with all its receptors and ligands expressed in normal tissues, is much more complex than the PD-1/PD-L1 axis.

The role of PVRIG as a potential regulator of HCC outcome however still remains to be proven in functional and animal model studies. Whereupon this validation, ICIs targeting PVRIG-NECTIN-2 interactions might be envisaged in HCC. For now, ICIs targeting the CTLA-4 and/or PD-1/PD-L1 axes have shown promising results in HCC [59,60]. Our observations suggest that the TIGIT/DNAM-1 axis, and notably the PVRIG/NECTIN-2 interaction, might be another interesting target for ICIs. This has never been considered for HCC yet. However, clinical trials evaluating the safety, tolerability, and clinical activity of COM701, an anti-PVRIG antibody, in monotherapy or in combination with a PD-1 inhibitor (NCT03667716) or an anti-TIGIT antibody (NCT04570839, BMS-986207), have been initiated in subjects with advanced solid tumors. Overall, these trials suggest that blocking the PVRIG/NECTIN-2 interaction to enhance anti-tumor cytotoxic function is a new therapeutic opportunity for the treatment of solid tumors [51]. Based on our work, our hypothesis is that immune cells, including cytotoxic cells, are recruited at the tumor site in *PVRIG* High tumors. They receive an activation signal and become activated, but they also receive negative signals from PVRIG, which is strongly engaged with NECTIN-2, which might limit anti-tumor activation. In these tumors, precluding the PVRIG/NECTIN-2 interaction might unleash a cytotoxic anti-tumor immune response (Figure 7) and improve the clinical outcome.

## 5. Conclusions

We showed that *PVRIG* expression is heterogeneous in HCC clinical samples and higher expression is associated with better DFS and OS and with strong, although non-optimal, local immune response. Our study displays several strengths: (i) its originality, (ii) a relatively large size of series, and (iii) the biological and clinical relevance of *PVRIG* expression. It also includes a few limitations: (i) its retrospective nature and associated biases, and (ii) the mRNA assessment, rather than protein, analysis on bulk tissue samples. Of course, analyses in larger series are warranted to confirm our observation, as well as functional analyses on HCC preclinical models to validate PVRIG’s role in HCC progression. However, and considering the inhibitory role of PVRIG, our results highlight the potential value of anti-PVRIG immunotherapies to enhance the anti-tumoral immune local response in HCC. Our data suggest that the microenvironment of *PVRIG* High samples (strong local cytotoxic immune response) might be more favorable for ICIs efficiency than that of *PVRIG* Low samples. However, the link between the target expression level and tumor response to ICIs remains unclear in many cancers. For example, in the CheckMate-040 trial dedicated to patients with HCC [18,19], the responses to nivolumab occurred irrespectively of the PD-L1 staining status. Clearly, the testing of anti-PVRIG ICIs in HCC is warranted at both pre-clinical and clinical levels. In this setting, analysis of pre-treated tumor samples will allow to assess *PVRIG* expression and to test if it can predict the therapeutic response. Overall, our study also highlights the complexity behind immune checkpoints function, revealing important roles for other potential candidates, such as here, with the members of the TIGIT/DNAM-1 axis.

## Figures and Tables

**Figure 1 cancers-15-00447-f001:**
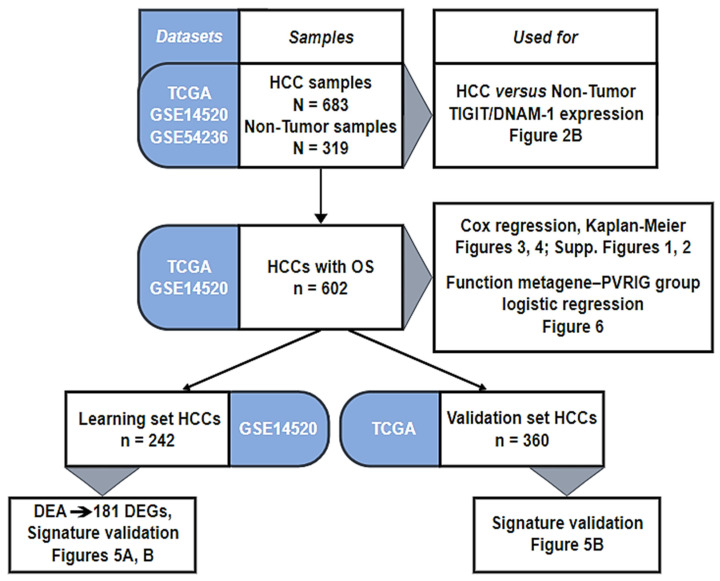
Analysis workflow.

**Figure 2 cancers-15-00447-f002:**
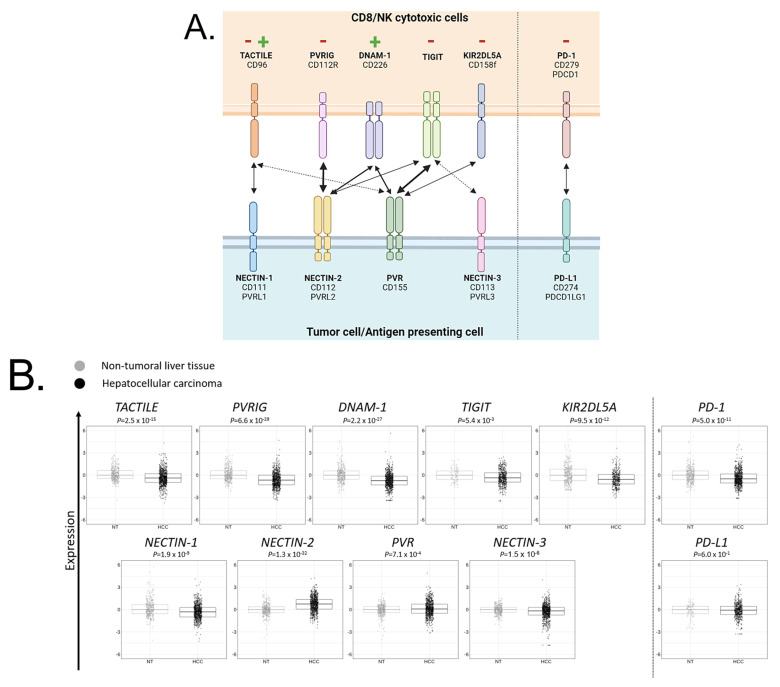
Members of the TIGIT/DNAM-1 axis and their respective gene expression in HCC and NT samples: (**A**) Members of the TIGIT/DNAM-1 axis and their respective ligands. TACTILE, PVRIG, DNAM-1, TIGIT, and KIR2DL5A are expressed on cytotoxic CD8^+^ T cells (CD8) and natural killer (NK) cells. NECTIN-1, NECTIN-2, NECTIN-3, and PVR are expressed on antigen-presenting cells (APC) or tumor cells. TACTILE binds NECTIN-1 and PVR with a stronger affinity for NECTIN-1. PVRIG binds NECTIN-2. DNAM-1 binds NECTIN-2 and PVR with equivalent affinity. TIGIT binds NECTIN-2, PVR, and NECTIN-3 with a stronger affinity for PVR and a weaker affinity for NECTIN-3. KIR2DL5A binds PVR. PVRIG, TIGIT, and KIR2DL5A contain immunoreceptor tyrosine-based inhibitory motif (ITIM) intracellular domain that triggers inhibitory signals, whereas DNAM-1 contains an immunoreceptor tyrosine-based activation motif (ITAM) intracellular domain that triggers activating signals. TACTILE signaling remains poorly described yet. PD-1/PD-L1 axis was added as a reference. Created with BioRender.com. (**B**) Gene expression level of each member of the TIGIT/DNAM-1 axis in NT and HCC liver tissues. *PD-1* and *PD-L1* were added as references.

**Figure 3 cancers-15-00447-f003:**
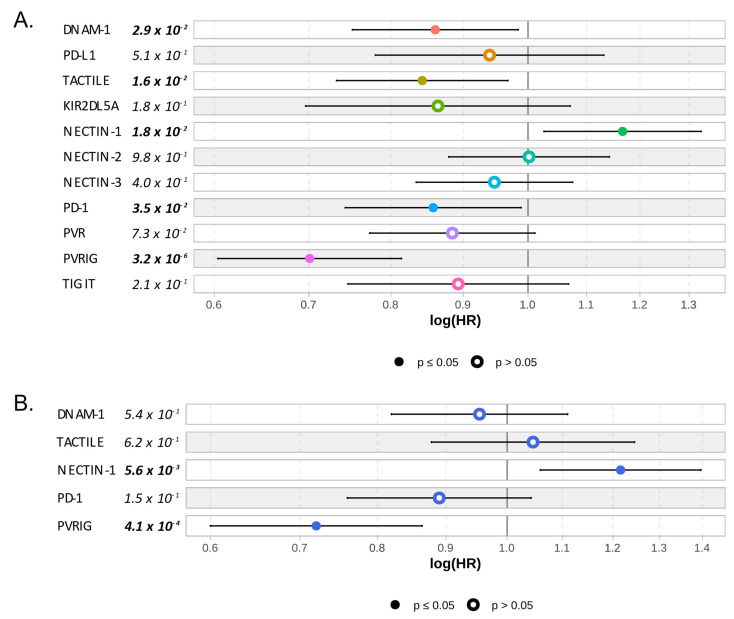
Correlation between gene expression of each member of the TIGIT/DNAM-1 axis and overall survival: (**A**) Univariate—one color by Cox regression—and (**B**) multivariate correlation analyses between each member of the TIGIT/DNAM-1 axis and OS. PD-1/PD-L1 axis was added for references. Cox model representation. Significant *p*-values (*p* ≤ 0.05) are written in bold and represented with plain circles.

**Figure 4 cancers-15-00447-f004:**
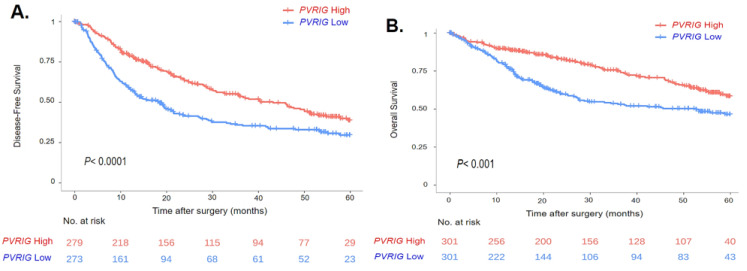
Survival curves in HCC patients based on *PVRIG* gene expression levels (High vs. Low): (**A**) Kaplan–Meier curve in HCC patients with DFS data (n = 552). (**B**) Kaplan–Meier curve in HCC patients with OS data (n= 602). The *p*-value indicates the log-rank test.

**Figure 5 cancers-15-00447-f005:**
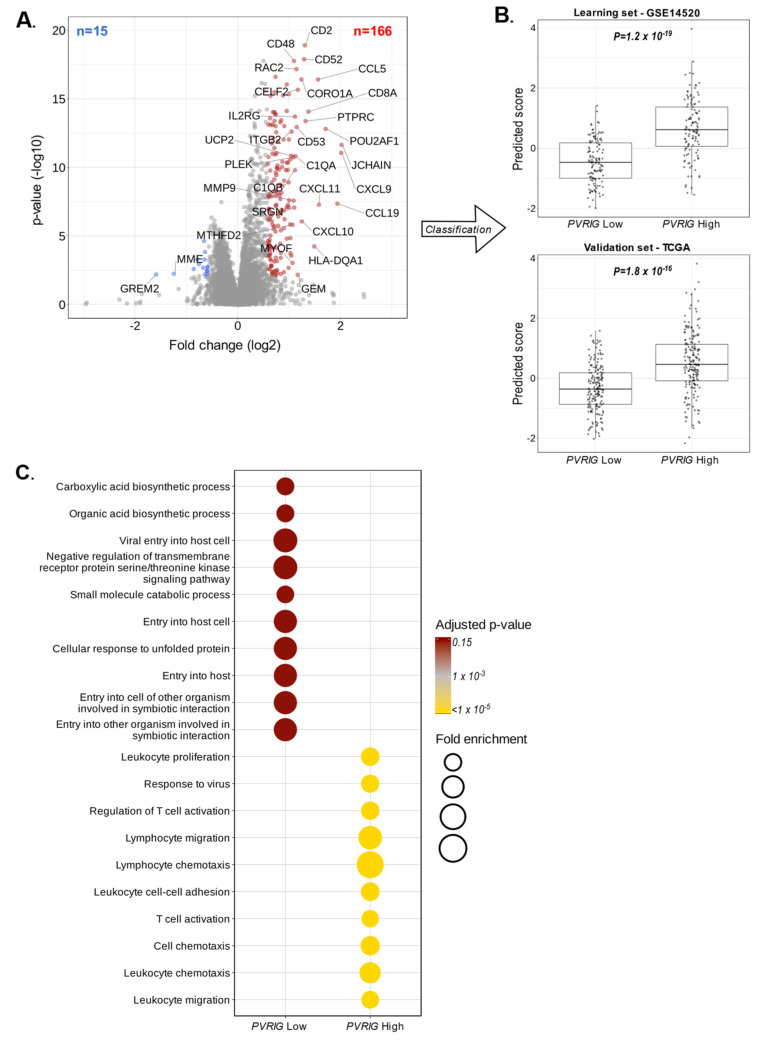
Correlations of *PVRIG* expression with biological processes: (**A**) Volcano plot of the 181 genes differentially expressed in the learning set (GSE14520, n = 242), with downregulated genes in *PVRIG* Low tumors (**left**, blue) and upregulated genes in *PVRIG* High tumors (**right**, red). *PVRIG* (x = 0.60; y = 38) is out of the scale. (**B**) Validation of robustness of the 181-gene list by comparing the metagene scores of *PVRIG* Low samples and *PVRIG* High samples using a Student *t*-test in the independent validation set (TCGA, n = 360). (**C**) Gene ontology (GO) enrichment analysis of 181 genes differentially expressed between *PVRIG* Low and High groups in the learning set (GSE14520, n = 242).

**Figure 6 cancers-15-00447-f006:**
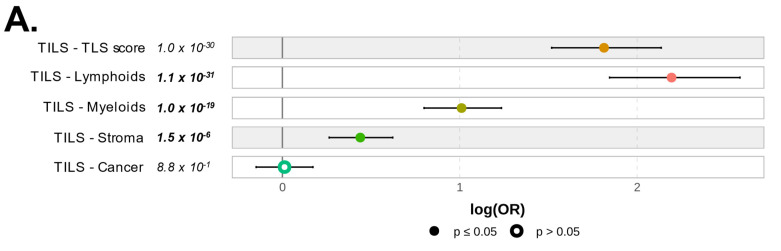
Correlation of *PVRIG* expression levels with immune and stromal features: (**A**) Forest plot representation of the correlations between *PVRIG* Low (**left**) and High (**right**) expression and the TILs score tool. (**B**) Forest plot representation of the correlations between *PVRIG* Low (left from the 0 value) and High (right from the 0 value) expression and the Bindea’s immunome (24 immune populations). (**C**) Forest plot representation of the correlations between *PVRIG* Low (**left**) and High (**right**) expression and immune functional signatures, including ICR, TIS, and TLS enrichment signatures, the immune cytolytic activity score, and the APM signature. (**D**) Forest plot representation of the correlations between *PVRIG* Low (**left**) and High (**right**) expression and CAFs subsets. The *p*-values are for the logit link test.

**Figure 7 cancers-15-00447-f007:**
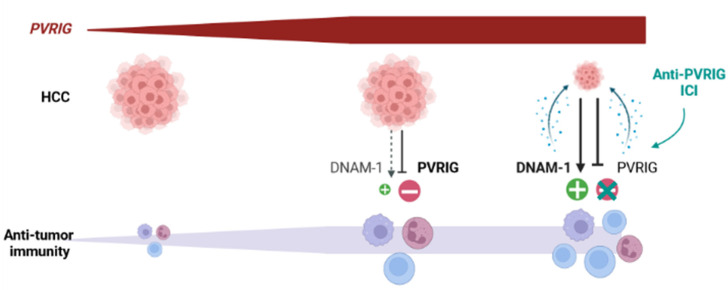
Hypothetical evolution of HCC after treatment with an anti-PVRIG in *PVRIG* High tumors: *PVRIG* High HCC are infiltrated with cytotoxic cells, whose activity is not fully unleashed. Our hypothesis is that anti-PVRIG treatment might alleviate this inhibitory signal (one among others) in innate and adaptive cytotoxic cells expressing PVRIG. This might favor anti-tumor activation and lead to better tumor regression and better clinical outcome.

**Table 1 cancers-15-00447-t001:** Clinicopathological characteristics of HCC samples (N = 683).

Characteristics		HCC
Age		
	Total (Median, [range] (years))	56 (16–90)
	≤50 years (n (%))	200 (33%)
	>50 years (n (%))	402 (67%)
Sex (n (%))		
	Female	165 (24%)
	Male	518 (76%)
TNM staging (n (%))		
	I	260 (46%)
	II	155 (28%)
	III	142 (25%)
	IV	4 (1%)
Pathological grade (n (%))		
	1	53 (15%)
	2	171 (48%)
	3	120 (34%)
	4	11 (3%)
AFP expression level (n (%))		
	≤300 ng/mL	281 (55%)
	>300 ng/mL	227 (45%)
HBV infection status (n (%))		
	Negative	244 (43%)
	Positive	321 (57%)
HCV infection status (n (%))		
	Negative	287 (84%)
	Positive	54 (16%)
Alcohol consumption (n (%))		
	Negative	226 (66%)
	Positive	115 (34%)
NAFLD (n (%))		
	Negative	321 (94%)
	Positive	20 (6%)
Hemochromatosis (n (%))		
	Negative	335 (98%)
	Positive	6 (2%)
Hoshida’s subtypes (n (%))		
	S1—Stromal	191 (31%)
	S2—Stemness—Angiogenic	132 (22%)
	S3—Differentiated	288 (47%)
DFS (Median (range), (months))		28 (1–121)
5 years DFS rate (95% CI, range)		34% (30–40)
OS (Median (range), (months))		70 (1–121)
5 years OS rate (95% CI, range)		53% (48–58)

**Table 2 cancers-15-00447-t002:** Correlations of *PVRIG* gene expression with clinicopathological characteristics.

Characteristics		*PVRIG* Groups		
		Low	High	*p*-Value
Age (n (%))	≤50 years	101 (34%)	99 (33%)	0.93
	>50 years	200 (66%)	202 (67%)	
Sex (n (%))	Female	73 (21%)	92 (27%)	2.9 × 10^−2^
	Male	269 (79%)	249 (73%)	
TNM staging (n (%))	I	117 (42%)	143 (51%)	0.1
	II	84 (30%)	71 (25%)	
	III	77 (27%)	65 (23%)	
	IV	3 (1%)	1 (0%)	
Pathological grade (n (%))	1	25 (14%)	28 (16%)	0.8
	2	86 (49%)	85 (48%)	
	3	59 (33%)	61 (34%)	
	4	7 (4%)	4 (2%)	
AFP expression level (n (%))	≤300 ng/mL	149 (58%)	132 (52%)	0.2
	>300 ng/mL	106 (42%)	121 (48%)	
HBV infection status (n (%))	Negative	124 (44%)	120 (43%)	0.9
	Positive	160 (56%)	161 (57%)	
HCV infection status (n (%))	Negative	146 (86%)	141 (83%)	0.5
	Positive	24 (14%)	30 (18%)	
Alcohol consumption (n (%))	Negative	113 (67%)	113 (66%)	1.0
	Positive	57 (34%)	58 (34%)	
NAFLD (n (%))	Negative	165 (97%)	156 (91%)	3.6 × 10^−2^
	Positive	5 (3%)	15 (9%)	
Hemochromatosis (n (%))	Negative	166 (98%)	169 (99%)	0.5
	Positive	4 (2%)	2 (1%)	
Hoshida’s subtypes (n (%))	S1—Stromal	64 (21%)	127 (41%)	1.9 × 10^−7^
	S2—Stemness—Angiogenic	73 (24%)	59 (19%)	
	S3—Differentiated	167 (55%)	121 (39%)	
DFS (Median (range), (months))		19 (1–115)	40 (1–121)	
5 years DFS rate (95% CI, range)		30% (24–37)	39% (32–46)	4.0 × 10^−5^
OS (Median (range), (months))		53 (1–114)	81 (1–121)	
5 years OS rate (95% CI, range)		47% (41–55)	58% (51–66)	1.2 × 10^−4^

**Table 3 cancers-15-00447-t003:** Univariate and multivariate prognostic analyses for DFS (n = 552).

Characteristics		Univariate		Multivariate	
		HR (95% CI)	*p*-Value	HR (95% CI)	*p*-Value
Age	≤50 years	1.0 (reference)	0.3	-	-
	>50 years	1.1 (0.9–1.4)		-	-
Sex	Female	1.0 (reference)	0.6	-	-
	Male	1.1 (0.8–1.4)		-	-
TNM staging	I	1.0 (reference)	1.9 × 10^−3^	1.0 (reference)	-
	II	2.0 (1.5–2.6)		2.0 (1.5–2.7)	4.6 × 10^−6^
	III	2.9 (2.2–3.9)		2.9 (2.1–3.9)	2.6 × 10^−12^
	IV	10.0 (3.2–33.0)		8.6 (2.7–28.0)	3.1 × 10^−4^
Pathological grade	1	1.0 (reference)	0.7	-	-
	2	1.3 (0.79–2.0)		-	-
	3	1.4 (0.8–2.2)		-	-
	4	1.3 (0.5–3.5)		-	-
AFP expression level	≤300 ng/mL	1.0 (reference)	0.2	-	-
	>300 ng/mL	0.9 (0.7–1.1)		-	-
HBV infection status	Negative	1.0 (reference)	1.8 × 10^−6^	1.0 (reference)	1.5 × 10^−4^
	Positive	0.6 (0.4–0.7)		0.6 (0.5–0.8)	
HCV infection status	Negative	1.0 (reference)	0.1	-	-
	Positive	1.4 (0.9–2.0)		-	-
Alcohol consumption	Negative	1.0 (reference)	0.9	-	-
	Positive	1.0 (0.7–1.4)		-	-
NAFLD	Negative	1.0 (reference)	0.8	-	-
	Positive	1.1 (0.6–2.1)		-	-
Hemochromatosis	Negative	1.0 (reference)	0.7	-	-
	Positive	0.7 (0.2–3)		-	-
Hoshida’s subtype	S1—Stromal	1.0 (reference)	0.1	-	-
	S2—Stemness—Angiogenic	1.2 (0.8–1.6)		-	-
	S3—Differentiated	0.8 (0.6–1.1)		-	-
PVRIG	Low	1.0 (reference)	5.1 × 10^−5^	1.0 (reference)	8.5 × 10^−4^
	High	0.6 (0.5–0.8)		0.7 (0.5–0.8)	

**Table 4 cancers-15-00447-t004:** Univariate and multivariate prognostic analyses for OS (n = 602).

Characteristics		Univariate		Multivariate	
		HR (95% CI)	*p*-Value	HR (95% CI)	*p*-Value
Age	≤50 years	1.0 (reference)	0.8	-	-
	>50 years	1.0 (0.8–1.3)		-	-
Sex	Female	1.0 (reference)	0.9	-	-
	Male	1.0 (0.7–1.4)		-	-
TNM staging	I	1.0 (reference)	1.5 × 10^−12^	1.0 (reference)	-
	II	1.7 (1.2–2.5)		1.7 (1.1–2.5)	1.9 × 10^−2^
	III	3.5 (2.5–5.0)		3.5 (2.4–5.2)	1.9 × 10^−10^
	IV	7.3 (2.3–23.0)		7.3 (2.2–24.0)	1.1 × 10^−3^
Pathological grade	1	1.0 (reference)	0.7	-	-
	2	1.3 (0.7–2.3)		-	-
	3	1.4 (0.8–2.5)		-	-
	4	1.5 (0.5–4.7)		-	-
AFP expression level	≤300 ng/mL	1.0 (reference)	0.1	-	-
	>300 ng/mL	1.3 (1.0–1.7)		-	-
HBV infection status	Negative	1.0 (reference)	1.5 × 10^−5^	1.0 (reference)	0.1
	Positive	0.5 (0.4–0.7)		0.7 (0.5–1.0)	
HCV infection status	Negative	1.0 (reference)	0.6	-	-
	Positive	1.1 (0.7–1.9)		-	-
Alcohol consumption	Negative	1.0 (reference)	0.7	-	-
	Positive	0.9 (0.6–1.4)		-	-
NAFLD	Negative	1.0 (reference)	0.3	-	-
	Positive	0.6 (0.2–1.5)		-	-
Hemochromatosis	Negative	1.0 (reference)	0.5	-	-
	Positive	0.5 (0.1–3.4)		-	-
Hoshida’s subtype	S1—Stromal	1.0 (reference)	4.2 × 10^−4^	1.0 (reference)	-
	S2—Stemness—Angiogenic	1.3 (0.9–1.8)		1.2 (0.8–1.8)	0.4
	S3—Differentiated	0.6 (0.5–0.9)		0.6 (0.4–0.9)	1.3 × 10^−2^
PVRIG	Low	1.0 (reference)	1.4 × 10^−4^	1.0 (reference)	9.1 × 10^−4^
	High	0.6 (0.5–0.8)		0.6 (0.4–0.8)	

## Data Availability

All datasets are publicly available, and references are described in Appendix A.

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
