# Peer review of "PVRIG Expression Is an Independent Prognostic Factor and a New Potential Target for Immunotherapy in Hepatocellular Carcinoma"

_cancers, 2023, doi:10.3390/cancers15020447_

Round 1

Reviewer 1 Report

I have gone through the manuscript. The topic is indeed interesting but authors have exclusively focused on clinical aspects of the patients. They have attempted to bring forth the role of TIGIT/DNAM role as regulator of HCC. However, it needs additional evidence related to animal model studies. there are limited studies related to  TIGIT/DNAM  role in carcinogenesis and metastasis, therefore, I appreciate the initiative of researchers to drill down deep into the dark areas of this signaling. Authors are encouraged to test these aspects in animal model studies by inoculation of  TIGIT/DNAM-silenced and overexpressing cells in mice to mechanistically validate the findings they have reported in this paper. 

Author Response

PVRIG Expression is an Independent Prognostic Factor and a New Potential Target for Immunotherapy in Hepatocellular Carcinoma

David J Birnbaum 1,2,3,†, Maelle Picard 1,2,†, Quentin Da Costa 1, Thomas Delayre 1,2,3, Pascal Finetti 1,2, Olivier Cabaud 1, Emilie Agavnian 4, Bernadette De Rauglaudre 1,2,5, Emilie Denicolaï 1,2, François Bertucci 1,2,6 and Emilie Mamessier 1,2,*

We are very grateful to the Editor and the reviewers for their comments. We would like to thank the Editor and the Reviewers for their time and helpful suggestions. The manuscript has been substantially reformatted and modified based on these suggestions and the conclusions have been amended. We apologize for the numerous typographical errors in the previous version and have now proofread the manuscript. We have reformatted the abstract according to the Editor’s suggestion. We have replied to all comments and have answered within the 15-days’ delay that was offered. Detailed responses to each of the reviewer’s comments are given below, in blue. We believe that the modifications that have been done has contributed to improve the general message and the overall interpretation of our findings. We hope that the Editors and the Reviewers will be pleased with the revised version and found it suitable for publication in Cancers.

Best regards,

Emilie Mamessier

Editor’s comments and Authors’ response:

The abstract was reformatted based on the following plan:

1) Background: Place the question addressed in a broad context and highlight the purpose of the study;2) Methods: Describe briefly the main methods or treatments applied. Include any relevant preregistration numbers, and species and strains of any animals used.

3) Results: Summarize the article's main findings;

4) Conclusion: Indicate the main conclusions or interpretations.

We apologize for not following these recommendations at first. 

Abstract:

Hepatocellular carcinoma (HCC) is a frequent and deadly cancer in need for new treatments. Immunotherapy has shown promising results in several solid tumors. The TIGIT/DNAM-1 axis gathers targets for new immune checkpoint inhibitors (ICIs). Here, we aimed at highlighting the potential of this axis as a new therapeutic option for HCC. For this, we built a large transcriptomic database of 683 HCC samples, clinically annotated, and 319 normal liver tissues. We interrogated this database for the transcriptomic expression of each member of the TIGIT/DNAM-1 axis and tested their prognostic value for survival. We then focused on the most discriminant one for these criteria, i.e. PVRIG, and analyzed the clinical characteristics, the disease-free and overall survivals, and biological pathways associated with PVRIG High tumors. Among all members of the TIGIT/DNAM-1 axis, PVRIG expression was higher in tumors than in normal liver, was heterogeneous across tumors, and was the only member with independent prognostic value for better survival. PVRIG High tumors were characterized by a higher lymphocytic infiltrate and enriched for signatures associated with tertiary-lymphoid structures and better anti-tumor immune response. These results suggest that patients with PVRIG High tumors might be good candidates for immune therapy involving ICIs, notably ICIs targeting the TIGIT/DNAM-1 axis. Further functional and clinical validation is urgently required.

Reviewer 1’s comments and Authors’ point-by-point response:

I have gone through the manuscript. The topic is indeed interesting but authors have exclusively focused on clinical aspects of the patients. They have attempted to bring forth the role of TIGIT/DNAM role as regulator of HCC. However, it needs additional evidence related to animal model studies. there are limited studies related to TIGIT/DNAM role in carcinogenesis and metastasis, therefore, I appreciate the initiative of researchers to drill down deep into the dark areas of this signaling. Authors are encouraged to test these aspects in animal model studies by inoculation of TIGIT/DNAM-silenced and overexpressing cells in mice to mechanistically validate the findings they have reported in this paper. 

A limit of our study is indeed to focus on the clinical aspects of the patients only. As a laboratory of precision oncology and translational medicine, this is what we do: we search for new therapeutic and/or prognostic targets for aggressive tumors. We fully agree with the need to perform functional assays and animal model studies to validate the role of the TIGIT/DNAM-1 axis as a regulator of HCC development. This is unfortunately not feasible within the 15-days’ delay that was granted to provide a revised version of the manuscript, not to mention that we unfortunately do not have the skills to perform HCC animal models.

However, based on the Reviewer’s comments,

1-  We have emphasized the need for functional assays and animal model studies to validate the role of the TIGIT/DNAM-1 axis as a potential regulator of HCC outcome in the discussion and conclusion sections.

On line 41-42, page 1, at the end of the Abstract, we have added the sentence: “Further functional and clinical validation is urgently required.”

On line 460-462, page 18, in the Discussion section, we have added the following sentences: “The role of PVRIG as a potential regulator of HCC outcome however still remains to be proven in functional and animal model studies. Whereupon this validation, ICIs targeting PVRIG-NECTIN2 interactions might be envisaged in HCC.”

On line 496-497, page 19, in the Conclusion section, the sentence “Of course, analysis of larger patient series is warranted to confirm our observation, as well as functional analysis of HCC preclinical models” has been replaced by the following sentence: ”Of course, analyses in larger patient series are warranted to confirm our observation, as well as functional analyses on HCC preclinical models to validate PVRIG’s role in HCC progression”.

2- We have taken additional precautions not to over-interpret the results.

In the title, line, 2 page 1, we have replaced “New Target for Immunotherapy” by “New Potential Target for Immunotherapy”.

In the Introduction section, line 76, page 2, we replaced “In this line, the TIGIT/DNAM-1 axis appears as an attractive option for the next generation of cancer immunotherapies” by “In this line, the TIGIT/DNAM-1 axis might be an option for the next generation of cancer immunotherapies”.

We also added the following sentence at the end of the Introduction section, line 101-104, page 3: “Altogether, these clinical trials suggest that the TIGIT/DNAM-1 axis is indeed a target for the upcoming generation of ICIs, and a new therapeutic opportunity for the treatment of several tumor types, potentially including HCC”.

In the Discussion section, line 405-407, page 17, we changed “Based on the analysis of expression of the TIGIT/DNAM-1 axis’ members in a large cohort of HCC clinical samples, our study highlights a pivotal role for PVRIG in the immune response against tumor cells” with the following sentence: “Based on the gene expression analysis of the TIGIT/DNAM-1 axis in a large cohort of HCC clinical samples, we showed that PVRIG is an independent biomarker predictive for favorable clinical outcome.”

3- We have provided additional references in line with a role for members of the TIGIT/DNAM1 axis during HCC progression. 

A very recent paper demonstrates that the blockade of TIGIT enhanced the antitumor activity of CD8+ T cells during the progression of HBV-related HCC in a spontaneous HCC mouse model. In this case, TIGIT blockade reinvigorated intrahepatic CD8+ T cells with increased TNF-α and IFN-γ production and an increased number of CD8+ T cells in tumors, thereby slowing the development of HCC in HBs-HepR mice. Interestingly, the blockade of PD-L1 did not slow HCC growth in HBs-HepR mice when administered alone, nor synergized with TIGIT blockade (Hepatology. 2022 Aug 8.doi: 10.1002/hep.32715. Online ahead of print.). In this case, TIGIT blockade alone provided better antitumor activity than anti-PD-L1. This recent finding provides robust data indicating that the TIGIT/DNAM-1 axis might be a better candidate than the PD1/PDL-1 axis in HCC (at least in HBV-related HCC).

This is now mentioned in the Introduction section (line 82-91, page 2), in which we have added the following sentence:

“Even though studies putting forward the role of TIGIT/DNAM role as a regulator of HCC are still particularly scarce in the literature, one recent study reports that the blockade of TIGIT enhanced the antitumor activity of CD8+ T cells during the progression of HBV-related HCC in a spontaneous HCC mouse model. Interestingly, the blockade of PD-L1 did not slow HCC growth in HBs-HepR mice when administered alone, nor synergized with TIGIT blockade. This recent finding provides robust data indicating that the TIGIT/DNAM-1 axis might be a better candidate than the PD1/PDL-1 axis in HCC (at least in HBV-related HCC). Additional examples along the same line can however be found for other tumor types.”

We would like to thank reviewer #1 as we believe that the addition of this new bibliographic reference indeed re-enforces the rational for looking at the TIGIT/DNAM-1 axis in HCC. 

Finally, extensive proofreading has been performed. We apologize for the numerous typographical and grammatical errors in the previous version.

Reviewer 2 Report

The present study investigated the clinical significance of PVRIG expression in the patients with hepatocellular carcinoma (HCC). The group of HCC patients with high PVRIG expression showed significantly worse prognosis. Various genes relating to immune response and inflammation up-regulated in the HCC patients with high PVRIG expression. In addition, the tumors with high PVRIG expression showed the significant infiltration of immune cells. The reviewer considers that the present study showed new evidences of the relationship among the PVRIG expression, the clinical outcome and the cytotoxic immune population. The reviewer would like to ask some queries to the authors as described below.

1.    In line 400-402, the authors described that these results suggested a higher anti-tumor immune response in PVRIG High tumor than PVRIG Low tumors that relies on both the adaptive and the innate immune systems. This suggests the immune response to HCC is sufficiently induced in the group with high PVRIG expression. On the other hand, the conclusion of the present study indicated that the about half of HCC might be susceptible to immune checkpoint inhibitors targeting the TIGIT-DNAM-1 axis. The reviewer wonders whether the PVRIG inhibitor treatment can contribute to improve the survival of patients with high PVRIG expression. Because the sufficient anti-tumor immune response is suggested to be induced in the patients with high PVRIG expression according to the present study, the possibility of inducing enhanced therapeutic effect by PVRIG inhibitor treatment is unclear. In addition, the influence of PVRIG expression to the therapeutic effect of PVRIG inhibitor is also unclear. Therefore, the reviewer considers that it is difficult to indicate this conclusion from the results of the present study. The reviewer would recommend to change some words in the part of conclusion, especially the description regarding the effect of PVRIG inhibitor.

2.    In main text, there are several typographical errors. Proofreading should be performed again.

Author Response

PVRIG Expression is an Independent Prognostic Factor and a New Potential Target for Immunotherapy in Hepatocellular Carcinoma

David J Birnbaum 1,2,3,†, Maelle Picard 1,2,†, Quentin Da Costa 1, Thomas Delayre 1,2,3, Pascal Finetti 1,2, Olivier Cabaud 1, Emilie Agavnian 4, Bernadette De Rauglaudre 1,2,5, Emilie Denicolaï 1,2, François Bertucci 1,2,6 and Emilie Mamessier 1,2,*

We are very grateful to the Editor and the reviewers for their comments. We would like to thank the Editor and the Reviewers for their time and helpful suggestions. The manuscript has been substantially reformatted and modified based on these suggestions and the conclusions have been amended. We apologize for the numerous typographical errors in the previous version and have now proofread the manuscript. We have reformatted the abstract according to the Editor’s suggestion. We have replied to all comments and have answered within the 15-days’ delay that was offered. Detailed responses to each of the reviewer’s comments are given below, in blue. We believe that the modifications that have been done has contributed to improve the general message and the overall interpretation of our findings. We hope that the Editors and the Reviewers will be pleased with the revised version and found it suitable for publication in Cancers.

Best regards,

Emilie Mamessier

Editor’s comments and Authors’ response:

The abstract was reformatted based on the following plan:

1) Background: Place the question addressed in a broad context and highlight the purpose of the study;2) Methods: Describe briefly the main methods or treatments applied. Include any relevant preregistration numbers, and species and strains of any animals used.

3) Results: Summarize the article's main findings;

4) Conclusion: Indicate the main conclusions or interpretations.

We apologize for not following these recommendations at first. 

Abstract:

Hepatocellular carcinoma (HCC) is a frequent and deadly cancer in need for new treatments. Immunotherapy has shown promising results in several solid tumors. The TIGIT/DNAM-1 axis gathers targets for new immune checkpoint inhibitors (ICIs). Here, we aimed at highlighting the potential of this axis as a new therapeutic option for HCC. For this, we built a large transcriptomic database of 683 HCC samples, clinically annotated, and 319 normal liver tissues. We interrogated this database for the transcriptomic expression of each member of the TIGIT/DNAM-1 axis and tested their prognostic value for survival. We then focused on the most discriminant one for these criteria, i.e. PVRIG, and analyzed the clinical characteristics, the disease-free and overall survivals, and biological pathways associated with PVRIG High tumors. Among all members of the TIGIT/DNAM-1 axis, PVRIG expression was higher in tumors than in normal liver, was heterogeneous across tumors, and was the only member with independent prognostic value for better survival. PVRIG High tumors were characterized by a higher lymphocytic infiltrate and enriched for signatures associated with tertiary-lymphoid structures and better anti-tumor immune response. These results suggest that patients with PVRIG High tumors might be good candidates for immune therapy involving ICIs, notably ICIs targeting the TIGIT/DNAM-1 axis. Further functional and clinical validation is urgently required.

Reviewer 2’s comments and Authors’ point-by-point response:

The present study investigated the clinical significance of PVRIG expression in the patients with hepatocellular carcinoma (HCC). The group of HCC patients with high PVRIG expression showed significantly worse prognosis. Various genes relating to immune response and inflammation up-regulated in the HCC patients with high PVRIG expression. In addition, the tumors with high PVRIG expression showed the significant infiltration of immune cells. The reviewer considers that the present study showed new evidences of the relationship among the PVRIG expression, the clinical outcome and the cytotoxic immune population. The reviewer would like to ask some queries to the authors as described below.

  1. In line 400-402, the authors described that these results suggested a higher anti-tumor immune response in PVRIG High tumor than PVRIG Low tumors that relies on both the adaptive and the innate immune systems. This suggests the immune response to HCC is sufficiently induced in the group with high PVRIG expression. On the other hand, the conclusion of the present study indicated that the about half of HCC might be susceptible to immune checkpoint inhibitors targeting the TIGIT-DNAM-1 axis. The reviewer wonders whether the PVRIG inhibitor treatment can contribute to improve the survival of patients with high PVRIG expression. Because the sufficient anti-tumor immune response is suggested to be induced in the patients with high PVRIG expression according to the present study, the possibility of inducing enhanced therapeutic effect by PVRIG inhibitor treatment is unclear.

We would like to thank the reviewer for this particularly relevant comment.

We have now:

  • Completed the Results section with data about immune activation and exhaustion markers. On line 379-388, page 17, we have added the following sentences:

“We then looked into the 166 genes up-regulated in the PVRIG High tumors (Table S4) and searched for signs of exhaustion, which would be coherent with tumor development despite cytotoxic infiltration and activity. Except for IDO1, none of the classical markers of exhaustion (PDCD1/PD-1, ENTPD1, LAG3, HAVCR2/TIM3, CTLA4, CD160, 2B4, BTLA, TIGIT, TOX, EOMES…) were detected [46]. Inversely, the absence of key molecules associated with highly efficient anti-tumor immune response (IFNG, IL12A and IL12B, IL18, TNF, GNLY, KLRC2, KLRC3,…) suggested that the cytotoxic potential of infiltrated immune cells was not totally unleashed despite expression of immune cell activation markers (GZMA, GZMB, GZMK, PRF1, CD8A, NKG7, STAT1, IRF1, TARP, KLRK1, PTPRC, CCL5, IL2RG, IL2RB, LCK, PIK3CD, CD69, CD52, CD53, TAP1…) (Table S4).

  • Improved the Discussion section (line 425-477, page 18) to explain this point more in detail.

“To our knowledge, the correlation between PVRIG expression and immune variables has never been explored further, and this is the first report characterizing the tumor immune microenvironment and biological pathways in PVRIG High tumors. The HBV and HCV infection status being equally distributed in PVRIG High or Low tumors, the background immune response due to the viral infection can be considered as equivalent. Compared to PVRIG Low tumors, several immune variables were enriched in PVRIG High tumors and suggested: (i) higher infiltration of immune cells; (ii) higher infiltration of both adaptive and innate immune cells, notably B cells, γδ T cells, T cells with a Th1 profile, cytotoxic CD8+ T cells and activated NK CD56dim cells; and (iii) higher cytotoxic activity. Altogether, these results suggested that PVRIG High tumors have a higher potential for anti-tumor immune response than PVRIG Low tumors. Nonetheless, despite this apparent anti-tumoral environment, the local immune cells did not prevent HCC occurrence and progression. The high expression of NECTIN-2, the ligand of PVRIG, which has also been reported by others in HCC tumors, might bridle anti-tumor response efficiency. Indeed, we did not find major evidence for exhaustion, as often observed in highly immunogenic tumors. The higher infiltration with both immunosuppressive and pro-tumoral CAF subsets in PVRIG High tumors was also in line with the hypothesis of a poor anti-tumoral response.

An apparent contradiction in our results lies in the association of PVRIG High expression, which can trigger a strong immunosuppressive signal, with better disease-free and overall survivals. We and others have previously reported similar results with expression of PD-L1 [53-55] and IDO1 inhibitory molecules [56]. In these cases, our explanation is that a very strong anti-tumoral immune response, associated with better clinical outcome, also activates an inhibitory feedback-loop, in the prospect of a return to homeostasis with minimum tissue damages. IFN-γ was in part responsible for the overexpression of PD-L1 and the initiation of the negative feedback-loop in these tumors [53]. Here, we did not report major signs of exhaustion, except for IDO1, and we did not detect strong IFN-γ production either, but there were evidences for general IFN and cytotoxic immune responses. One hypothesis might be that the PVRIG/NECTIN-2 axis is a new negative feedback-loop, either proper to the tumor liver microenvironment or associated with a specific context of immune response. Indeed, the TIGIT-DNAM-1 axis, with all its receptors and ligands expressed in normal tissues, is much more complex than the PD-1/PD-L1 axis.

The role of PVRIG as a potential regulator of HCC outcome however still remains to be proven in functional and animal model studies. Whereupon this validation, ICIs targeting PVRIG-NECTIN-2 interactions might be envisaged in HCC. For now, ICIs targeting the CTLA-4 and/or PD-1/PD-L1 axes have shown promising results in HCC [48, 49]. Our observations suggest that the TIGIT/DNAM-1 axis, and notably the PVRIG/NECTIN-2 interaction, might be another interesting target for ICIs. This has never been considered for HCC yet. But clinical trials evaluating the safety, tolerability and clinical activity of COM701, an anti-PVRIG antibody, in monotherapy or in combination with a PD-1 inhibitor (NCT03667716) or an anti-TIGIT antibody (NCT04570839, BMS-986207), have been initiated in subjects with advanced solid tumors. Overall, these trials suggest that blocking the PVRIG/NECTIN-2 interaction to enhance anti-tumor cytotoxic function is a new therapeutic opportunity for the treatment of solid tumors [51]. Based on our work, our hypothesis is that immune cells, including cytotoxic cells, are recruited at the tumor site in PVRIG High tumors. They receive activation signal and get activated, but they also receive negative signals from PVRIG, which is strongly engaged with NECTIN-2, which might limit anti-tumor activation. In these tumors, precluding the PVRIG/NECTIN-2 interaction might unleash cytotoxic anti-tumor immune response (Figure 7) and improve the clinical outcome.”

  • Provided a figure in the discussion section to expose our hypothesis (page 19). 

Figure 7. Hypothetical evolution of HCC after treatment with an anti-PVRIG in PVRIG High tumors. PVRIG High HCC are infiltrated with cytotoxic cells, whose activity is not fully unleashed. Our hypothesis is that anti-PVRIG treatment might alleviate this inhibitory signal (one among others) in innate and adaptive cytotoxic cells expressing PVRIG. This might favor anti-tumor activation and lead to better tumor regression and better clinical outcome.

  1. In addition, the influence of PVRIG expression to the therapeutic effect of PVRIG inhibitor is also unclear. Therefore, the reviewer considers that it is difficult to indicate this conclusion from the results of the present study. The reviewer would recommend to change some words in the part of conclusion, especially the description regarding the effect of PVRIG inhibitor.

We have now mentioned that our interpretation was hypothetical and make sure not to over-interpret our data throughout the manuscript.

In the Conclusion section, we have added the following sentences, line 500-508, page 19: “Our data suggest that the microenvironment of PVRIG High samples (strong local cytotoxic immune response) might be more favorable for ICIs efficiency than that of PVRIG Low samples. However, the link between the target expression level and tumor response to ICIs remains unclear in many cancers. For example, in the CheckMate-040 trial dedicated to patients with HCC [18, 19], the responses to nivolumab occurred irrespectively of the PD-L1 staining status. Clearly, the testing of anti-PVRIG ICI in HCC is warranted at both pre-clinical and clinical levels. In this setting, analysis of pre-treated tumor samples will allow to assess PVRIG expression and to test if it can predict the therapeutic response.”

  1. In main text, there are several typographical errors. Proofreading should be performed again.

Extensive proofreading has been performed. We apologize for the numerous typographical and grammatical errors in the previous version.

We would like to thank reviewer #2 as we believe that the discussion section is now clearer, which contributes to re-enforces our findings. 

Round 2

Reviewer 1 Report

Looks good now